# Feasibility-Usability Study of a Tablet App Adapted Specifically for Persons with Cognitive Impairment—SMART4MD (Support Monitoring and Reminder Technology for Mild Dementia)

**DOI:** 10.3390/ijerph17186816

**Published:** 2020-09-18

**Authors:** Maria Quintana, Peter Anderberg, Johan Sanmartin Berglund, Joakim Frögren, Neus Cano, Selim Cellek, Jufen Zhang, Maite Garolera

**Affiliations:** 1Brain, Cognition and Behavior: Clinical Research, Consorci Sanitari de Terrassa, 08227 Terrasa, Spain; mquintana@cst.cat (M.Q.); ncanom@cst.cat (N.C.); 2Department of Health, Blekinge Institute of Technology, SE-371 79 Karlskrona, Sweden; peter.anderberg@bth.se (P.A.); johan.sanmartin.berglund@bth.se (J.S.B.); joakim.frogren@med.lu.se (J.F.); 3Faculty of Health, Education, Medicine & Social Care, Anglia Ruskin University, Chelmsford CM1 1SQ, Essex, UK; selim.cellek@anglia.ac.uk; 4Clinical Trials Unit, Anglia Ruskin University, Chelmsford CM1 1SQ, Essex, UK; Jufen.Zhang@anglia.ac.uk; 5Neuropsychology Unit, Hospital de Terrassa, Consorci Sanitari de Terrassa, 08227 Terrassa, Spain

**Keywords:** dementia, monitoring, e-health, feasibility study

## Abstract

Population ageing within Europe has major social and economic consequences. One of the most devastating conditions that predominantly affects older people is dementia. The SMART4MD (Support Monitoring and Reminder Technology for Mild Dementia) project aims to develop and test a health application specifically designed for people with mild dementia. The aim of this feasibility study was to evaluate the design of the SMART4MD protocol, including recruitment, screening, baseline examination and data management, and to test the SMART4MD application for functionality and usability before utilization in a full-scale study. The feasibility study tested the protocol and the app in Spain and Sweden. A total of nineteen persons with cognitive impairment, and their informal caregivers, individually performed a task-based usability test of the SMART4MD app model in a clinical environment, followed by four-week testing of the app in the home environment. By employing a user-centered design approach, the SMART4MD application proved to be an adequate and feasible interface for an eHealth intervention. In the final usability test, a score of 81% satisfied users was obtained. The possibility to test the application in all the procedures included in the study generated important information on how to present the technology to the users and how to improve these procedures.

## 1. Introduction

The number of older adults in the population as well as their proportion within the total population are increasing in many parts of the world. In Europe, population aging has had and will continue to have major social and economic consequences. This is a fundamentally positive development where the added life span is of great benefit for both individuals and for society. Many older people are able to support themselves and continue to make important contributions to society. Yet, the risk for the individual to contract noncommunicable diseases and disability increases with age. This may impair the ability of an individual to live his/her life in the way that is desired. It also exerts pressure on health services and support systems for older people.

Cognitive conditions such as mild cognitive impairment (MCI) and dementia constitute a group of chronic diseases that predominantly affect older people [1]. The general increase in life span causes the number of people with MCI [2] to increase; the number of people with dementia in Europe alone is estimated to reach 13.4 million by 2030.

Touchscreen tablets are increasingly used to support people with dementia. Tablets are used for psychosocial interventions, life support, social living and leisure activities. Autonomous use is possible and group use can promote interpersonal relationships [3]. Different studies [4,5] offer a progressive look at how tablets can be applied to improve communication between persons living with dementia and their caregivers.

Health-oriented applications on mobile units such as smartphones and PDAs, known as mHealth applications, can be useful to both support elderly people who have cognitive impairment and their informal caregivers [6,7] and to improve their quality of life [8,9]. However, some studies suggest that a prerequisite for older persons with cognitive impairment to start using computer-based technology is that it offers individual customization according to personal preferences [10,11]. It is important to be aware of the heterogeneity in technology use abilities [12,13,14]. Involving end users in design and development of technology is especially important for the development of usable technology [15,16,17].

Many devices and digital apps to support the day-to-day living activities at home for people with neurocognitive disorders have been developed in recent years. However, most solutions have focused on cognitive impairment training or support and safety for people living alone. In the on-going Horizon 2020 project SMART4MD (Support, Monitoring And Reminder Technology for older persons with Mild Dementia) [18], a model health-oriented app has been developed through a user-centered process involving stakeholders in three European countries and with an emphasis on customization to allow for the various needs of older persons with cognitive impairment and their informal caregivers.

Many devices and digital apps to support the day-to-day living activities at home for people with neurocognitive disorders have been developed in recent years. However, most solutions have focused on cognitive impairment training or support and safety for people living alone. Most of them are single apps or devices, but few have been integrated into a single solution or product [19,20].

The implementation of eHealth apps in this MCI population, however, is sometimes challenging to their cognitive and age characteristics. Some of the key factors in the successful implementation of eHealth apps are usability and feasibility. Usability is defined by the International Standards Organization as “the effectiveness, efficiency, and satisfaction with which specified users can achieve goals in particular environments” [21].

A feasibility study is an assessment of the practicality of a proposed project or system. It is used to determine whether an intervention is appropriate for further testing and to evaluate its effectiveness [22]. The feasibility is based on the direct observation of the user’s experiences, as well as the difficulties with which it is derived during or after its implementation [23].

Previous studies have been published, such as the one by Castilla et al. [24], in order to study the usability and acceptability of a prototype for elderly care, in an attempt to explore the Information and Communication Technologies (ICTs) design needs of users with MCI. The main contribution of this study consists of exploring the usability needs of users with MCI on ICT systems and providing some usability recommendations for designing interfaces for this kind of user. Other accessibility studies in MCI users such as that of Haesner et al. [25] showed why subjects with MCI needed more time and were more likely to make mistakes when using a web platform than those subjects without cognitive impairment. There is also the study carried out by Hattink et al. [26], which evaluates the usability and usefulness of an online portal for patients with dementia and their carers.

The aim of this SMART4MD feasibility-usability study was to evaluate the design of the SMART4MD study protocol, including recruitment, screening, baseline examination and data management, and to test the SMART4MD application for functionality and usability before initiating a full-scale RCT (Randomised Controlled Trial). The purpose of conducting the feasibility study in two different countries was to examine whether any linguistic and/or cultural differences had an impact on the perception of the application.

## 2. Method

### 2.1. Design

The design of this feasibility-usability study, in which a clinical usability testing session is combined with a four-week test period in the participant’s home environment, as have other authors such as Sheehan and Lucero [27]. According to these authors, their method is a suitable way of combining the validation of changes (based on earlier iterations of user feedback) with an assessment of specific features. The method was appropriate as earlier design iterations had already been done and needed validation, whilst other features were newly developed and had not received any user feedback. Furthermore, this feasibility-usability study is carried out before the implementation effort of the full-scale SMART4MD study, as recommended in the scientific literature [23].

Our feasibility-usability study has been divided into two phases or parts, which are described below:

#### 2.1.1. Phase 1

The first part consisted of a one-time introduction to the application and task-based testing in a clinical setting. This part consisted of a classic usability test [28], where the SMART4MD app was used in a controlled environment. The assessment method used was a task analysis [29], applied individually, where the user performed several predefined tasks in order to obtain quantitative and qualitative data. As feasibility is associated with users’ actual experiences with an innovation in a specific context, in vivo usability testing is particularly relevant, as indicated by authors such as Hermes et al. [23].

Usability testing as described by Goodman at al. [30] is based on letting potential users explore and test a product or service through the performance of a number of tasks. According to this method, the tasks should be constructed in such a way that they let the users explore the core functionality of the product/service in a realistic way, i.e., based on the intended users and their potential usage. According to this user testing method, users should not be given a guided introduction to the app before the testing session, since this might bias their opinions and result in less valuable feedback. Instead, they should be introduced to the app individually through the tasks they are asked to perform. They should not be given clues except when really necessary, and after each task has been performed, questions should be asked about their experience of solving that task. In order to get valuable feedback, it is recommended that 6–10 users should be employed for the testing so that each task is performed by a minimum of five users [30].

#### 2.1.2. Phase 2

The second part consisted of a four-week test period of the tablet and app in the dyad’s home environment, followed by a user evaluation.

### 2.2. Participants

#### 2.2.1. Eligibility Criteria

##### Inclusion Criteria

A participant will be eligible for inclusion in this trial only if all of the following criteria are met:-Participants score 20 to 28 points on the Mini Mental State Examination (MMSE) whether or not a diagnosed neurodegenerative disease is present;-A professional assessment of the patient’s own experience of memory problems over a substantial period of time (more than 6 months);-Participants are older than 55 years;-Participants are home care recipients;-Participants have an informal carer;-Participants take prescribed medication and are in charge of their own medication use;-Participants have no specific conditions reducing their physical ability to use the app, for example, visual, hearing, or motor impairments.

##### Exclusion Criteria (Persons with Mild Cognitive Impairment Only)

A participant will not be eligible for inclusion in this study if any of the following criteria apply:-Participants have a terminal illness with less than 3 years of expected survival;-Participants score above 11 on the Geriatric Depression Scale (GDS-15) or have another known significant cause of disease as an explanation for cognitive impairment such as abuse and other psychiatric diagnoses such as bipolar disorders, schizophrenia, and developmental disorders.

#### 2.2.2. Sample Size

The aim of the screening procedure was to identify 10 dyads at each site (*n* = 20 dyads in total, or 40 individuals) who fulfilled the eligibility criteria of the feasibility study. The dyad would be comprised of the person with dementia and their caregiver. The reasons for including such a seemingly large number of participants were several. Most importantly, the SMART4MD project is aimed at a quite broad spectrum of people, from 55 years of age upwards, with variations both in cognitive ability and in familiarity with tablet computers and apps. In other words, the variables were many and it was therefore considered important to receive feedback from participants with varying ages, cognitive abilities, and previous technical knowledge. Furthermore, since the SMART4MD app contains numerous sections, it could not be expected that all users should be able to perform all the tasks, and therefore more participants than the recommended 6–10 were considered necessary. The aim was to receive feedback from at least five users per task and site.

#### 2.2.3. Ethical Considerations and Informed Consent

Ethical approval for this study was granted by the regional ethical review boards at each participating site ensuring full compliance with all research and legislative regulations in the respective countries. The ethical code is BTH: SMART4MD dnr 2016/470, approved EPN Lund 2016-06-21. CST: project identification code: 02-16-107-029, approved by Consorci Sanitari de Terrassa Ethics Committee on 25 April 2016

At the start of the potential participants’ visit to the clinical site, information about the study, based on the Participation Information Sheet, was orally presented to the dyads to ensure that they fully understood the meaning of their participation in the study. If agreeing to participate, the participants (both the People with Mild Cognitive Impairment (PwMCI) and the caregiver) were asked to sign an informed consent document. The original signed document was kept by the clinical site and a copy of the signed document was given to the dyad.

All confidential data were stored by the research team in a safe place, as required by Data Protection Directive 95/46/EC, the Personal Data Act (Sweden), and the Organic Law 15/1999 on Personal Data Protection dated December 13 (Spain). All confidential digital material, such as study participant files, were also stored in secure folders on a computer only accessible by the principal investigators at each site and the clinical study staff involved in the study. Non-digital material, such as notes from the usability testing, were kept in safety lockers when not being analyzed.

#### 2.2.4. Sample Characteristics

The BTH dyads in the baseline sample had an average age of 77.5 years. The PwMCI sample had a mean age of 77 years; 67% of the sample were men (*n* = 6), and these subjects had an average MMSE of 25.2 and an average GDS of 2.0. Five of the nine PwMCI had used a smartphone or tablet almost every day during the three months preceding the baseline visit. The carers had a mean age of 68 years, of which 55% were men (*n* = 5). The relationship with the PwMCI was mainly spouses/common-law partners (*n* = 7), followed by child (*n* = 1) and other (*n* = 1).

The CST dyads in the baseline sample had an average age of 79.9 years. The PwMCI sample had a mean age of 80 years. Fifty percent of the sample were women (*n* = 5), and these subjects had an average MMSE of 23.4 and an average GDS of 2.9. Seven of the ten PwMCIs had never used a smartphone or tablet. The carers had a mean age of 64 years, of which 70% were men (*n* = 7). The relationship with the PwMCI was mainly spouses/common-law partners (*n* = 5), followed by child (*n* = 4) and other (*n* = 1).

### 2.3. Materials

#### 2.3.1. Variables and Measuring Instrument

This study includes the variables and instruments from Table 1.

##### Screening Variables

The screening consisted of two instruments, the Mini Mental State Examination (MMSE) [31] and the Geriatric Depression Scale (GDS) [32], and a number of demographic questions considered relevant for the study. The MMSE was used to briefly assess the cognitive functions of the PwMCI. It includes questions about orientation, attention, recall, and language ability. It is also used to estimate the severity and progression of cognitive impairment and to follow the course of cognitive changes in an individual over time. To be included in this study, individuals must score between 20 and 28 points on the scale. The use of an MMSE cutoff value of 28 is not common and has some risks but has been used in other studies [33]. O’Bryant et al. [34] showed that an MMSE cutoff score of 28 gave the best sensitivity and specificity for detecting mild dementia in a population with self-reported memory complaints.

The GDS was used as an exclusion criterion, screening for depression in PwMCI. If participants score above 11, they will be excluded from the study. As people who are physically ill and living with mild to moderate dementia have short attention spans and/or feel easily fatigued, we used the short form of the GDS (GDS-15) consisting of 15 questions. The GDS is commonly used as a routine part of a comprehensive geriatric assessment. The grid sets a range of 0 to 4 as “normal”, 5 to 8 as “mildly depressed”, 9 to 11 as “moderately depressed”, and 12 to 15 as “severely depressed”.

##### Usability Testing Variables

In the usability testing, the following quantitative metrics on efficacy and efficiency were collected, these variables were selected based on the scientific literature [23].

-Ability to complete the task independently: The ability to complete the task independently was measured when the task was completed autonomously, that is, the user did not require the help of the evaluator.-Time to complete the tasks (seconds).-Need for guidance (by caregiver or clinical test leader) to complete the task.

In addition, subjective metrics were collected, including questions after each completed task.

The qualitative metrics were the following:-Sections where the user struggled.-Difficulties experienced by the user when completing the task.-Features missing that would substantially help according to the dyad.

##### User Data from SMART4MD Application

The only user data assessed in the feasibility study was the amount of SIM data/user/month.

##### Weekly Calls

The dyads were offered the opportunity to receive a weekly call from the research team during the four-week test period. All dyads agreed to this. The purpose of this contact was to enquire whether the dyad needed assistance with the tablet or the application. The following series of questions were directed to the dyads at the weekly calls:Have you had any problems with the SMART4MD platform during the week?Do you have any questions regarding the use of the SMART4MD platform?

All contacts with the users during the test period were logged and all questions and problems they encountered registered.

##### User Evaluation Variables

The user evaluation was conducted through a structured interview with the PwMCI and carer separately. The interviews were divided into two sections. The first section consisted of an evaluation specifically tailored to the study. It was based on the most significant quality attributes for general user satisfaction among users of similar health information technology [35] and on principles that provide the foundation for web accessibility according to WCAG 2.0 conformance requirements). This section covered domains such as accessibility, safety/trustability, perceivability, understandability, and empowerment:Accessibility: I find the application easily accessible.Safety and Trustability: I feel that I can trust the application and that it is safe to use.Perceivability: I find it easy to understand how to operate the application.Understandability: I am able to understand all the information presented in the application.Empowerment: I feel that the application gave me a better control over my daily situation.

Specifically, the following questions were asked to the PwMCI:How satisfied are you with the application’s possibility to support you?How well does the application fulfill your expectations?Imagine a perfect application for this task. How far away from it is the application you are using today?I find the application easily accessible for different kinds of users.I feel that I can trust the application and that it is safe to use.I find it easy to understand how to operate the application.I am able to understand all the information presented in the application.I feel that the application gave me better control over my daily situation.

The following questions were asked to the carers:How satisfied are you with the application’s possibility to support your relative (and you)?How well does the application fulfill your expectations?Imagine a perfect application to support your relative (and you). How far away from that is the application that your relative and you are using today?I find the application easily accessible for different kinds of users.I feel that my relative and I can trust the application and that it is safe to use.I find it easy for my relative (and me) to understand how to operate the application.My relative (and I) are able to understand all the information presented in the application.I feel that the application gave my relative a better control over his/her daily situation.

The alternatives the users could give to all the above were the following: 1 = Strongly disagree; 2 = Disagree.; 3 = Neither agree nor disagree; 4 = Agree; 5 = Strongly agree.

The second part of the user evaluation consisted of a standard form for evaluating usability: the System Usability Scale (SUS) [36]. SUS provides a “quick and dirty”, reliable tool for measuring the usability. It consists of a 10 item questionnaire with five response options for respondents; from Strongly agree to Strongly disagree.

##### General Usability and Utility Feedback Data

Throughout the course of the feasibility-usability study, the most relevant issues regarding usability and utility described by the dyads were continuously registered, summarized and analyzed. Descriptive statistics were used to summarize the data. This information was then provided to the developers of the app in order to improve the usability of the application for the full-scale RCT.

#### 2.3.2. Hardware

Tablets with 7” screen and Android 6.0 software and compatible hardware were used for the project. At CST, the tablet used was a Leotec Pulsar Q36, with a 7” screen, and at BTH a Lenovo Tab 7 with a 7” screen.

At BTH, a Telenor Micro SIM was used and data was limited to 10 GB/month/user. At CST, a Vodafone Global SIM was used and data limited to 750 MB/month/user. The Vodafone SIM card also provided an M2M platform.

In addition, in order to record usability training sessions in audio, an external voice recorder which was not connected to the internet was used. These audio recordings were transferred to and stored in secure folders on a password-protected computer which was only accessible by the principal investigators at each site and the clinical study staff involved in the study.

#### 2.3.3. Software

SMART4MD is a general electronic health app that has been adapted specifically for MCI through a structured process involving the participation of PwMCI and their informal carers and health care professionals. Focus groups were conducted with PwMCI and their informal carers, and interviews were conducted with health care professionals. The app can run both on tablet and mobile phone devices adopting the Android operating system but is specifically optimized for tablet devices as opposed to smaller screen mobile phone devices.

The core functionalities of the app are based on reminders (medication, appointments with health care providers, and meeting up with family and friends), cognitive supporting activities (clock, calendar, brain games, and photos) and optional status and health information sharing with family and informal carers (including mood and specific health problems such as headaches). The app allows the participant to share health information with other people of their choice (relatives or research team) by email. The participant can select what kind of information he/she can share and to whom he/she wants to send this information.

An important feature of SMART4MD is its personalization facility: main users (PwMCI and informal carers) will be able to switch on/off or change various features and information-sharing possibilities.

The app is intended to be used daily at home, mainly by the PwMCI themselves, with the help of their informal carers when needed.

In the present study, in order to test a newer and more complete version of the app, the dyads were offered an update of the app two weeks into the test period. For practical reasons, the updating of the app was performed with slight differences at the two sites. At BTH, the dyads were invited to receive the update at the same location where phase I of the usability testing had taken place. At CST, the test leader updated the app by paying the dyads a visit in their homes.

#### 2.3.4. Task Description

The tasks that were requested of the participants in the user testing were the following:

##### Reminders for activities

Add a reminder that you today in 30 minutes’ time from now should “call theatre and change tickets”. Set the reminder to remind you 10 min before the phone call.Add a reminder that you tomorrow at 10 am will have a general health checkup.Add a reminder USING A VOICE COMMAND that you tomorrow at 2 pm will have a general health checkup.Check the reminders you have added for today and tomorrow.

##### People I know

Add the professional staff (CST) and the professional staff (BTH) as a contact in the “People I know” section of the app.

##### Games and Resources

Go to the Games and Resources section and check what is included there

##### Medicine reminders

In the section “My Health”, add a medicine reminder to take the medicine Plavix, 2 times per day at 9 AM, and 6 PM starting from today.Check medicines that have been added.Delete medicines that have been added.

##### My Health: Symptoms

In the “My Health” section of the app, add that you have experienced a symptom. The symptom I want you to add that you have experienced is headache. Write that it started today and that the severity is 4 on a scale from 1 to 10.Check the symptoms that have been added.

##### Share with others

Now, I want you to share your symptom with others. The person I want you to share your symptom with is the professional staff (CST)/the professional staff (BTH) who has the email address xxxxx@hotmail.com.

##### About dementia

Go to the About dementia section and check the information presented.Click on the different chapters and skim through them.

##### General tasks

Turn on the tablet.Start the SMART4MD application.

### 2.4. Procedure

#### 2.4.1. Sites, Recruitment, Flowchart, and Timeline

This study was conducted in two countries: in Sweden by the Blekinge Institute of Technology (Swedish initials—BTH), and in Spain by the Consorci Sanitari de Terrassa (CST).

Site BTH is located in Karlskrona (Blekinge). At BTH, the recruitment process mainly consisted of contacting persons who had previously shown an interest in participating in the study. A pre-screening interview via telephone was conducted with these potential participants. The pre-screening comprised questions on age, community dwelling, the existence of an informal caregiver, and self-rated memory.

Site CST is located in Terrassa (Barcelona). At CST, potential participants were identified by reviewing primary care service patient databases. Of the seven primary care centres associated with CST, the Primary Care Centre in Castellbisbal was selected for the feasibility study. However, due to the difficulty of finding suitable candidates in Castellbisbal, subjects from the Sant Jordi Day Hospital for Cognitive Impairment were recruited. Of the pre-selected participants who were contacted by telephone, 72.5% declined to participate in the study. The main reason given was that the caregiver saw difficulties in the use of the application by people with mild cognitive impairment (PwMCI).

All the potential participants who met the pre-screening criteria were sent a Participation Information Sheet with information about the study. The participants were scheduled for a screening/baseline visit >24 h after informally agreeing to participate in the study. This was done to allow them a period of reflection to consider their participation. If the potential participants changed their mind during this period, they were encouraged to contact the research team by phone or mail.

Figure 1 presents the flowchart for this study.

The selected dyads were then to proceed to the baseline examination and phases I and II of the feasibility-usability study.

#### 2.4.2. General Tablets Settings

The tablets were prepared in such a way that external apps were deleted from the home screen in order to make interaction with the SMART4MD app simpler. Furthermore, a number of changes in the settings were made in order to improve accessibility and usability, such as horizontal/vertical lock, turning off screen lock, etc. A light blue background image was downloaded and set as the screen background. Specifically, the tablets were prepared as follows:Charge tabletFixed portrait orientationSettings > Display > When device is rotated > Stay in current orientationSleep after 30 min of inactivitySettings > Display > Sleep > 30 minRemove screen lockSettings > Security > Screen lock > NoneDesktop is blank(SMART4MD launcher only on screen, all other apps/widgets removed from desktop)Remove non-essential appsDisable notifications from all other apps (including games)

#### 2.4.3. Installation of SMART4MD App

The SMART4MD app was sent as an apk file, first from developer to the technicians at CST and BTH, who were responsible for installing the app on each of the tablets, and then from the technicians to the different email address of the individual tablets from which the app was installed. An individual email account was set up on each tablet following the system smart4md.NNNN@, where “NNNN” was the PwMCI’s ID number from the screening.

#### 2.4.4. SMART4MD App Settings

The cognitive games that are used together with the app were installed individually for each tablet through Google Play. After the games had been installed, the icons for the games were deleted so that the games could be accessed solely from within the app. A change of settings was also carried out to make the app more accessible and to increase usability.

#### 2.4.5. Introduction and User Testing of Tablet and SMART4MD App (Phase I)

The introduction to the tablet and SMART4MD app and user testing were carried out at the BTH facilities (Sweden) by a PhD student in Applied Health Technology with an educational background in Cognitive Science and Interaction Design.

In Spain, the introduction to the tablet and SMART4MD application was performed by a PhD student in neuropsychology with extensive experience in new technologies and people living with dementia, specifically at the Castellbisbal Primary Care Centre and the Sant Jordi Cognitive Impairment Hospital (Terrassa).

The participants were first contacted by phone and scheduled for the introduction session. They were also sent a reminder letter to ensure that there were no misunderstandings concerning the time. When a dyad arrived at the introduction session, they were first given general information about the setup of the feasibility-usability study as a whole followed by more specific information about the introductory meeting. They were also informed and asked to give their consent to having the session audio recorded.

This first phase of the feasibility-usability testing was then done individually with first the PwMCI and then the carer. The experimenter had the role of directing the training/testing session of the application. First, there would be an explanation of the application. Then the subject or the caregiver would be shown how the application was used, and both of them would then be asked to carry out some tasks. While one person in the dyad did the usability testing, the other was asked to wait outside. Since not all participants had time to complete all the tasks within the timeframe, the tasks were divided up so that each task would be tested by at least five participants. The most central tasks, which were part of the “reminder section” of the app and which had to be explained to all users, were tested by more participants than the other tasks. All the tasks that were requested were described in the previous section, Task description.

#### 2.4.6. Delivery of the Tablet with SMART4MD App for Use at Home for Four Weeks (Start Phase II)

When both persons in the dyad had completed the usability testing in June 2017, the test leader met with them together and answered any questions that they had about the tablet, app, or the study. The dyads were then informed about the rights and regulations concerning their app and tablet use and were asked to sign an agreement form for their usage. The tablets were then handed over to the dyads. The users received the initial version of the SMART4MD app so that they could start immediately. About two weeks after using the initial version, all the participants received an updated version of the app, which they were able to use for the remaining two weeks of the test. Around the end of October and the beginning of November, a complementary round of testing and user assessment was conducted after addressing all programming challenges.

From the moment of the delivery of the tablet, the second phase of this study began. The second part consisted of a four-week test period of the tablet and app in the dyad’s home environment, followed by a user evaluation, as described below. The dyads were informed that they were free to contact the test leader during the test period should they have any questions. They were also offered weekly calls from the test leader and, midway through the test period, met the test leader to receive an update of the app.

#### 2.4.7. User Evaluation Session (End Phase II)

When the four-week test period was over, the dyads were invited to the clinical sites again for a user evaluation. The user evaluation was conducted through a structured interview with the PwMCI and carer separately. The interviews were divided into two sections. The first section consisted of an evaluation specifically tailored to the study (see section variables and measuring instrument for more detail).

The second part of the user evaluation consisted of a standard form for evaluating usability: the System Usability Scale (SUS). After completing the user evaluation, the CST dyads returned their tablets and ended their participation in the SMART4MD feasibility study. However, the BTH dyads were offered to keep their tablets and to continue evaluating the usability of future updates of the SMART4MD app.

## 3. Results

### 3.1. Usability and Feasibility Results

The first phase of the feasibility-usability study took place between 8–16 June 2017 at CST and BTH. This session consisted of a short introduction followed by task-based individual usability testing with first the PwMCI and then the carer. The session took about 2–2.5 h/dyad. Tablet pens to facilitate usage were provided for the participants during the usability testing session, which were highly appreciated by some. Results of the user performance of tasks included in the usability-feasibility testing are presented in Table 2.

There was a big difference in how familiar the various participants were with similar technology. Several PwMCI (62%, *n* = 13) said that they were curious to know and understand how tablets and smartphones worked. However, when asking their children or grandchildren for an explanation, they also said this was given in a way that was too fast and hard to follow for them. For this reason, exploring the app at their own speed with the possibility of asking questions when they did not understand was considered by the majority to be a very positive way to approach this technology.

A few PwMCI (43%, *n* = 9) who were not familiar with tablets or smartphones were very insecure when approaching the tablet and app. They explicitly said that, rather than exploring the tablet and app by themselves, they would have preferred to be shown clearly where to click. Others approached the technology with curiosity, but with extreme care and showing their fear of clicking somewhere without first receiving approval from the test leader, seemingly thinking that something might go wrong or get broken if they clicked somewhere without approval.

Participants who had previous smartphone/table knowledge were generally self-confident and had high self-esteem with regards to the app, sometimes complaining about certain features on the table or regarding the app and its functionality, comparing both to systems they were currently using.

The cognitive games included in the app were appreciated by all participants. Information about dementia was found to be well-written and informative. However, there were parts where users struggled when solving tasks. A detailed list of this can be found in Table 3.

This section may be divided into subheadings. It should provide a concise and precise description of the experimental results, their interpretation, and the experimental conclusions that can be drawn.

Finally, in reference to the data used in the SIM card, in this usability-feasibility study, the average usage was 641 MB/user/month.

### 3.2. User Evaluation Results

User evaluation in the form of a structured interview was conducted individually with all participants and took place at CST between 7–19 July and at BTH between 10–20 July. As previously mentioned, two quantitative measures were included in the user evaluation. First, the assessment of the different attributes previously described: accessibility, safety/trustability, perceivability, understandability, and empowerment. The second quantitative measure was the SUS.

User evaluation was analyzed in the following way: if a respondent had a minimum total score of 60% (15 out of 25) or more, he or she was considered to be satisfied with the application. Thirty out of thirty-seven (81%) of the users had a score of 15 or higher. The average score was 19. According to the standard operating protocol for SMART4MD, the feasibility study was to be considered successfully completed if at least 15 of the 20 dyads (75%) were satisfied with use of the application. In other words, the feasibility study was from this point of view considered successfully completed. However, as some new functionalities and updates had not yet been fully tested, a decision was made to have a complementary round of usability testing and evaluation as soon as the remaining parts of the app were fully functional. Table 4 presents the data of the user evaluation scores, specifically, the summation of the assessment of each of the assessed attributes of the SMART4MD app is presented, in addition to the evaluations of the different attributes.

### 3.3. Lessons Learned

Based on the user feedback from the feasibility-usability study and dialogue with the developer, a number of ideas for increased usability of the SMART4MD app were developed and integrated into the SMART4MD software. For a detailed list see Table 5.

Hardware—Tablets

The 7” tablets used in the feasibility study were considered surprisingly small by several participants. While some experienced the size as too small, others preferred a smaller tablet compared to a larger one since they saw it as more portable. In other words, there was no consensus about whether a 7” tablet is suitable for the target group or not. In the full-scale RCT, a 7” Vodafone tablet (Tab mini) will be used by all sites.

Software

Before phase I of the feasibility-usability study, the tablets were prepared in such a way that any external apps were deleted from the home screen in order to make interaction with the SMART4MD app simpler for the target group. Furthermore, a number of settings were changed in order to improve accessibility and usability, including horizontal/vertical lock, turning off screen lock, etc. A light blue background image was downloaded and set as the screen background.

Email accounts had to be set up manually on each tablet as the SMART4MD app needed to be installed through an APK file sent to these addresses. The games—unique for each site—then had to be installed manually through Google Play on each tablet. However, relying on manually created individual email accounts on each tablet is neither safe nor efficient. In addition, changing the setting manually for each tablet in the way that it was done is unnecessarily time-consuming. For these reasons, an MDM platform will be used in the full pilot of the SMART4MD study for installation of the SMART4MD app and administering the settings.

Introduction to the technology

A thorough and accessible introduction to the tablet and app was found to be crucial to facilitate and motivate the study participants to use the tablet and app independently in their home environment. In the full-scale RCT, the guidelines for the clinical study staff need to stress the importance of adjusting the level of the introduction to the person in the dyad with the least familiarity with similar technology.

Tablet pens

In addition, tablet pens found to ease the use of the tablets for some users will be available during the introduction in the full-scale RCT and will be recommended for the users to acquire.

A paper-based manual as a support and complement to the introduction

A paper-based manual was seen by several dyads (*n* = 10) in the feasibility study as desirable as a complement to the introduction. For the full-scale RCT, a thorough and accessible paper-based manual has therefore been developed and will be provided to all dyads in the intervention group. The manual has been developed with an emphasis on the parts of the tablet and app where users in the feasibility study experienced difficulties.

Recruitment

Recruitment was successful as evidenced by the acceptance rate of the participants when the study was explained to them. This ratio is higher in the case of Sweden (47.12%) than in the case of Spain (27.5%).

## 4. Discussion

The aim of the feasibility-usability study was to evaluate the setup of the study and the usability of the SMART4MD application before starting a full-scale RCT.

Although there has been an exponential increase in the number of eHealth apps, the number of studies that have been published that report the results of usability testing on these apps has not increased at an equivalent rate. The number of digital health applications that publish their usability evaluation results remains only a small fraction [35]. Therefore, this study represents a contribution to the literature, which can be used for other studies similar to this one. SMART4MD has been developed from an existing general health management application in a process involving the structured participation of PwMCI, their informal caregivers, and clinicians. The adoption of SMART4MD by older people faces the challenge of this age group’s relative unfamiliarity with digital devices and services. However, this challenge can also be seen as an opportunity. The aim of this SMART4MD feasibility study was to evaluate the design of the SMART4MD study protocol, including recruitment, screening, baseline examination, and data management, and to test the SMART4MD application for functionality and usability before utilization in a full-scale study. This feasibility study was conducted by two clinical partners and countries: Blekinge Institute of Technology, BTH (Sweden), and Consorci Sanitari de Terrassa, CST (Spain). The purpose of conducting the feasibility study in two different countries was to examine whether any linguistic and cultural differences impacted on the perception of the application, considering that the full pilot will be conducted at five sites in four different countries. This feasibility and usability testing was conducted in two phases. Phase I consisted of a one-time introduction and exercises with the application. Phase II consisted of a usage of the application in the participant’s home environment during a period of four weeks followed by a user evaluation. The design of this feasibility study is similar to the feasibility study conducted by Sheehan and Lucero [27].

Recruitment was successful as evidenced by the acceptance rate of the participants when the study was explained to them. This ratio is higher in the case of Sweden (47.12%) than in the case of Spain (27.5%). This difference in the recruitment ratio is due to the strategy used in each site. While in BTH the recruitment process mainly consisted of contacting persons who had previously shown interest in participating in the study and sending out by post information on the study to a group between 80 and 90 years old. Recruitment took place locally to avoid dropout due to distance and long travels. In the case of CST, the databases were reviewed to make a pre-selection of subjects that met the inclusion criteria. Afterwards, the participants were contacted. Mainly, the first telephone contact to invite to participate in this study was made with the main caregiver. Many caregivers decided that the subject could have difficulties in using the tablet, mainly due to the little or no experience they had in using this type of technology, which was the reason why they refused to participate in the study.

The evaluation protocol was adequate, both in the administration time and in the use of the electronic data collection notebook.

Some of the lessons learned in this study that avoid problems of feasibility and usability are as follows: a thorough introduction to the app was appreciated; a paper-based manual was desirable as a complement to introduction; and tablet pens are recommended to be had available during the introduction.

In this study, SUS did not prove to be suitable for this target group as many of the respondents’ answers were contradictory. There are indications that this might be due to the fact that the questions are sometimes asked with a double negative, something that might be hard for older participants with cognitive impairment to understand correctly. The user evaluation was therefore solely based on the degree of agreement of the PwMCI and caregiver individually with the statements evaluated with the questions that were designed for this study (see section measures to see the questions asked).

The overall experience with the SMART4MD app was positive. Rigorous testing was carried out with representatives from the user group and a final score of 81% satisfied users was obtained. In the Standard Operating Procedure of the SMART4MD project, it was stated that the criteria for success was that 75% of the users should be satisfied with the application before the project could advance to the clinical trial. From the results of this feasibility study, it is therefore concluded that the final version can be tested in a randomized controlled trial.

Findings from this study are in line with previous research in this area [25]. We found that older users and users with dementia are able and willing to utilize ICT solutions and that at least some of them are capable of using the technology involved.

In other usability studies similar to our study, elderly users face a greater number of usability problems than young users [37], and their ICT experience differs not only in terms of their success rate, but also in terms of emotional factors that should be included as an important part of their experience [38]. Our results indicate that familiarity with similar technology varies strongly among participants. Several PWDs expressed that they were curious to get to know and understand how tablets and smartphones worked but felt that when asking their children or grandchildren for an introduction, it was done in a way that was too fast and hard to follow for them. For that reason, exploring the app at their own speed and having the possibility to ask questions when they did not understand, was by a majority seen to be a very positive way of approaching this technology. A few PWDs who were not familiar with tablets or smartphones were very unsecure when approaching the tablet and app. Preliminary studies showed [39] that the use of touchscreen technologies was frequent also in PWD but that the use of specific apps or software to support memory was reduced in PWD and caregivers and they face age barriers for the use of these apps. However, the majority think that the use of smartphones or tablets is helpful for memory and this highlights the presence of a gap between the perceived potential and the actual use of these technologies. Moreover, limited prior exposure to similar technology affects both ability and self-esteem when confronted with the model app, and evaluating usability with the target group using standard forms in usability testing requires precautions.

Finally, data from the use of SIM data was relevant for deciding what would be the limit to be established in the clinical trial. In this usability-feasibility study, the average usage was 641 MB/user/month. However, in the feasibility study, several updates to the app were made and games were downloaded using the SIM. In the clinical trial, where the amount of SIM data was limited to 60 MB/user/month, Wi-Fi will be used for these updates and downloads.

## 5. Conclusions

In this research work we proposed an approach called SMART4MD, which combined of a number of functions that distinguish it from other comparable products and services, making it the kind of innovative non-pharmacological intervention that experts in the field have called for [40]. Our feasibility and usability study provides several recommendations that must be taken into account before starting the clinical trial of the project and that can be used in the future in similar studies. Specifically, we have brought forward the following insights from this feasibility-usability study to the full pilot study, and recommendations for other similar studies can be considered.

Introduction to the technology◦That ICT knowledge and how used the aim group is to this kind of technology varies a lot.◦That less exposure to similar technology, effects both ability and self-esteem and need to be taken into consideration when introducing app.◦That a thorough introduction to the app was appreciated by all and especially among those with less of a habit of using similar technology.◦A paper-based manual as a complement to the introduction to the app was asked for by some users in order to remember how to use the app.

Content of the SMART4MD app◦That the possibility of filling in symptoms as they appear (to be able to remember symptom development later) was seen as a very positive feature of the app.◦That the information section about dementia was seen as very informative.◦That cognitive games were perceived mostly positively by the users, especially since other functionalities were a bit limited in the feasibility study version of the app.◦That the occurring “bugs” in the app, especially in the reminder section, had a strongly negative impact on the impression of that section and on overall user satisfaction.◦That a few individuals with less cognitive impairment and a greater awareness and knowledge of what is possible to achieve with technology today, found this technology very limited and not useful.◦That individuals 90+ find this technology challenging due to their overall bodily function such as swollen fingers, and really needed to see a clear purpose with this technology in order to find it worthy to spend valuable time exploring the technology.

App development◦That more “common ground” between developers of technology and end users would make the process easier and would require less iterations.◦That the necessity of having a thoroughly tested, and thereby stable app (without bugs) to hand out to users in order for them to dare to explore it fully and to (hopefully) appreciate it better.

Therefore, all these insights represent a significant contribution to the development and validation of ICT solutions for subjects with cognitive impairment.

The occurrence of dementia and mild cognitive impairment is a common problem across Europe but its management varies between countries and even regions. Variations in the commitment of member states to national dementia strategies naturally translates into variations in clinical practice, through differences in both funding and priority given to dementia in different countries. More generally, healthcare systems vary widely between member states and regions, with differing emphases placed on home, community, and primary and secondary care. These differences are further amplified by varying practices in the reimbursement of healthcare costs between private healthcare insurers and public authorities and by different levels of funding available for healthcare. As a result, the support and treatment available for PwMCI varies widely across the EU.

The results of this study need to be considered in the context of several limitations. On the one hand, more participants could have been included in this feasibility and usability study. Furthermore, the study carried out in two countries, Sweden and Spain, and could have been extended to other participants in the SMART4MD project such as Belgium or the Czech Republic. Therefore, future studies could expand the number of participating countries, or propose the use of the app at home for a longer period of time. On the other hand, other feasibility and usability methodologies could also be considered than those used in this study. In conclusion, the possibility to test the SMART4MD application in all the procedures included in the study gave several important insights into how to introduce the technology to the users and how to improve these procedures, specifically in subjects with cognitive impairment. Therefore, the recommendations of this study could be used in similar studies: eHealth app, technology in dementia and cognitive impairment, for example.

## Figures and Tables

**Figure 1 ijerph-17-06816-f001:**
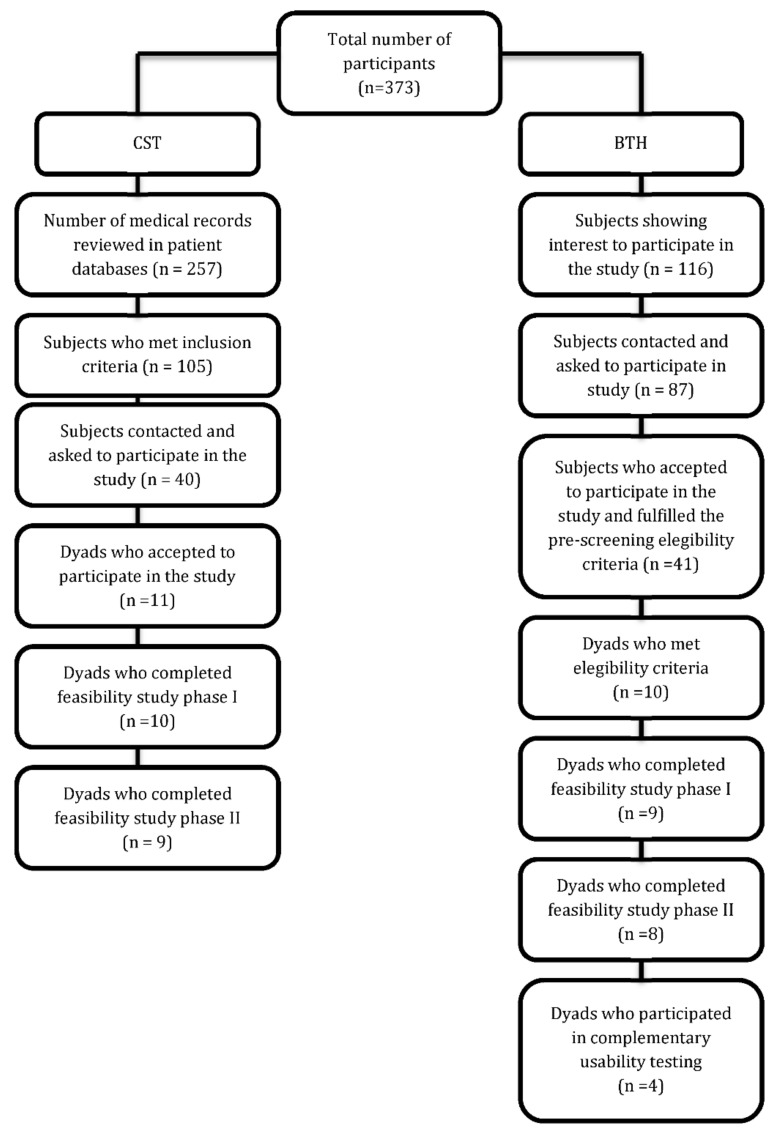
Flowchart for the feasibility study.

**Table 1 ijerph-17-06816-t001:** List of variables and instruments included in this study.

Measure	PwMCI	Carer	Place and Conditions	Period
MMSE	X		BTH facilities or CST facilities	Screening
GDS	X		BTH facilities or CST facilities	Screening
Usability testing	X	X	BTH facilities or CST facilities	Phase I
User data SIM	X	X	Home	During phase II (four weeks)
Weekly calls	X	X	Home	During phase II (four weeks)
User evaluation	X	X	BTH facilities or CST facilities	At the end of phase II
SUS	X	X	BTH facilities or CST facilities	At the end of phase II

**Table 2 ijerph-17-06816-t002:** User performance of task included in the usability-feasibility testing.

PWD				
Tasks	Succeded by Themselves	Succeded with Some Guidance	Succeded with Detailed Explicit Instructions	Average Time to Complete Task (s)
Turning the tablet on and off	15%	62%	23%	58
Starting the SMART4MD app	21%	71%	8%	44
Add a to do reminder	9%	91%	0%	400
Add an appointment reminder	0%	89%	11%	526
Add an appointment reminder using voice	0%	78%	22%	384
Check reminders for today and tomorrow	25%	38%	38%	113
Add a contact to ”People I know”	18%	82%	0%	259
Add a symptom	33%	67%	0%	173
**Carer**				
**Tasks**	**Succeded by Themselves**	**Succeded with Some Guidance**	**Succeded with Detailed Explicit Instructions**	**Average Time to Complete Task(s)**
Turning the tablet on and off	73%	20%	7%	19
Starting the SMART4MD app	91%	9%	0%	6
Add a to do reminder	20%	80%	0%	174
Add an appointment reminder	57%	43%	0%	174
Add an appointment reminder using voice	27%	64%	9%	211
Check reminders for today and tomorrow	73%	18%	9%	43
Add a contact to ”People I know”	47%	53%	0%	117
Add a symptom	67%	33%	0%	93

**Table 3 ijerph-17-06816-t003:** Parts where users struggled when solving tasks in the usability testing.

Tasks	Parts Where Users Struggled
Turning on or off the tablet	Pressing the start button long enough
Starting the SMART4MD app	Identifying and associating the SMART4MD logo within the app
Add a to do reminder	Where to tap to be able to start writing text; pressing the keys too long when writing; where to tap to select the day; where to tap to minimize keyboard; noticing the “Remind me in…” button (and therefore not filling in a reminder); understanding how to set the time correctly using the clock; finding the “space” key; finding the “OK” to confirm the day; finding the “erase” key to delete text
Add an appointment reminder	Same issues as listed in “Add a to do reminder” + understanding the difference between “to do” and “appointments” in this context (and therefore choosing “to do” instead of “appointments”); writing all info “freely” under “location” and not under the intended category; finding a key to create a new row (“Enter” key)
Add an appointment reminder using voice	Some of the issues listed in “Add a to do reminder” and “Add an appointment reminder” + finding the microphone button; assuming that a “smiley” icon on a keyboard was the microphone option; turning the microphone on and off; not understanding that all of the parts (i.e., not the calendar or the clock) functioned with a voice command
Check reminders for today and tomorrow	Hard to identify the button to check tomorrow’s reminders
Add a contact to “People I know”	Where to tap to be able to start writing text; where to tap to fill in the next field; finding the “space” key; what button to press to write with capital letters solely; understanding the necessity of pressing “Save” to confirm; finding the “erase” key to delete text; finding the numbers on the keyboard; where to click to add a photo
Add a symptom	Where to tap to be able to start writing text; where to tap to fill in the next field; Where to tap to select the date; how to move the severity scale indicator

**Table 4 ijerph-17-06816-t004:** User evaluation scores.

	Number Tested	Minimum Score	Maximum Score	Mean Score	SD
All	37	11	25	17.8	3.5
PwMCI CST	10	12	25	18.6	3.8
PwMCI BTH	8	12	21	16.3	3.4
PwMCI All	18	12	25	17.6	3.7
Carer CST	10	11	22	17.9	3.2
Carer BTH	9	13	24	18.2	3.7
Carer All	19	11	24	18.1	3.4
Accessibility	37	2	5	3.51	0.99
Safety	37	1	5	3.78	1.84
Perceivability	37	2	5	3.57	0.96
Understandability	37	2	5	3.76	0.96
Empowerment	37	1	5	3.19	1.10

**Table 5 ijerph-17-06816-t005:** Ideas for increased usability based on feedback from usability testing.

App Section	Tasks	Ideas for Increased Usability
General	Turning on/off the tablet	Instructions in paper-based manual
General	Starting the SMART4MD app	Larger icon; Instructions in paper-based manual
General	Main menu of app	Change “Resources” to “News’” Change icon for “symptom” to an image of two persons; make “People I Know” RED (#BF0C0C), Games and News YELLOW (#FFDA1A) and “Personalize my app” ORANGE (#FF7200); Have “Menu” written in capital letters and make it more button-like
My reminders	Add a “to do” reminder	Erase icons and make field outline stronger; add option “all day”; add option “30 min before” in “Remind me in…”; add pop-up “Are you sure you want to delete this activity?” (Y/N)
My reminders/My Health	Add an appointment reminder/Add an appointment reminder using voice	Erase icons and make field outline stronger; change “appointments” to “medical appointments”; add option “30 min before” in “Remind me in…”; add pop-up “Are you sure you want to delete this activity?”
My reminders	Check reminders for today and tomorrow	Instructions in paper-based manual
My reminders	Reminder goes off	Make reminder pop-up bigger, should cover 1/4 of the screen; reminder pop-up should remain on screen until interacted with; when clicking on reminder pop-up, a bigger pop-up should show all entered info; reminder should have snooze function 5 min; default sound twice as loud and twice as long as today
People I know	Add a contact to “People I know”	Erase icons and make field outline stronger; option to mark any user as “ICE” and make appear first in list; possibility to add hyphen and space when adding phone number; have choice to “Take photo” or “Add photo from gallery” when clicking “add a photo” + make first option possible; instead of “description”, have the option “Who is this person?”; add field for email; make photos keep their original proportions
Games and Resources	Check out “Games and Resources”	Change “Resources” to “News”
My Health	Add a symptom	Make field outline stronger; place assistive text inside field always; add optional text square for short description of symptoms
Share with others	Share symptoms with others/Share (health-related) appointments with others	Change “Invite people” to “Who do you want to share with?”; Change parameters in “Enter Sharing Details” section to “Name (of the person to share with)”, “email address” and optional “message”
About dementia	Check out “About dementia”	Increase text size
Personalize my app	Check out “Personalize my app”	Set 24 h clock as default; add bigger additional text size; possibility to change between three different signals, and listen to them while changing; make it possible to hear volume level as you change it;

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
