# Peer review of "Feasibility-Usability Study of a Tablet App Adapted Specifically for Persons with Cognitive Impairment—SMART4MD (Support Monitoring and Reminder Technology for Mild Dementia)"

_ijerph, 2020, doi:10.3390/ijerph17186816_

Round 1
Reviewer 1 Report
This study is a relevant work. Some points that could be observed to improve this contribution are:
a) Title: probably the authors could consider to have a more clear title
SMART4MD: Support Monitoring and Remind Technology for Mild Dementia
Mention the device (tablet) in the title could restrict search in the near future, if, for example, other equipment could utilize the application. The main goal of the contribution is the support monitoring and remind technology for mild dementia.
b) Abstract: The acronymous SMART4MD must be stated, because again the abstract could be print alone in other context.
c) Keywords: Are these keywords contributing to full understanding the work contribution? (Dementia, e-health, feasibility study) Monitoring, for instance, is not considered.
d) Document organization:
d1) A final paragraph in the Introduction with the text structure could help the reader to understand better the contribution.
d2) From page 3 until 8 several sub-sections exists without numbers, which does not help the understanding for the reader. It is necessary to build his/her own structure link;
d3) Page 4 appear the wrong text bellow Figure 1 "Figure 2. 6.2. SIM cards."
d4) From page 9 until 13 it is very the same aspects comment in d2.
d5) Related work: It is important to mention other efforts, as it is stated in conclusion section as an affirmative text. This could be conceived before/after presenting the present contribution
d6) Conclusions: two mentions could be considered:
1> Re-write this section to better understanding the contribution, example is the first sentence:
"SMART4MD’s combination of a number of functions distinguishes it from other comparable 476 products and services, making it the kind of innovative non-pharmacological intervention that 477 experts in the field have called for [17]."
Probably consider: In this research work we proposed an approach, called as SMART4MD, which....
2> Consider to insert Future Works, which could be done from you or other research groups.
d7) e-Health, software engineering and hardware: this is a general comment to contribute to the authors to think about. This work has a nice contributions in its basis, or high level target to monitoring and provide a tool for Mild Dementia. Software engineering and hardware are not "the most important aspects". These could change, but the proposed idea no.
Author Response
Dear Reviewer
First of all, thank you for reviewing our paper. Next, we respond to each of your comments and suggestions:
a) Title: probably the authors could consider to have a more clear title
SMART4MD: Support Monitoring and Remind Technology for Mild Dementia
Mention the device (tablet) in the title could restrict search in the near future, if, for example, other equipment could utilize the application. The main goal of the contribution is the support monitoring and remind technology for mild dementia.
We think your suggestion is very interesting, but at the moment SMART4MD app has been designed and tested to be used on tablets, instead of other devices, so we think it is relevant to include it in the title
b) Abstract: The acronymous SMART4MD must be stated, because again the abstract could be print alone in other context.
Thank you for your comment. We have included the meaning of the acronym in the abstract
c) Keywords: Are these keywords contributing to full understanding the work contribution? (Dementia, e-health, feasibility study) Monitoring, for instance, is not considered.
Thanks for your suggestion. We have included monitoring as a keyword
d) Document organization
d1) A final paragraph in the Introduction with the text structure could help the reader to understand better the contribution.
Thanks for your suggestion. The introduction has been expanded and reorganized (see paper).
d2) From page 3 until 8 several sub-sections exists without numbers, which does not help the understanding for the reader. It is necessary to build his/her own structure link;
Thanks for your suggestion. The organization of the paper has been partially modified in order to facilitate its reading (see paper).
d3) Page 4 appear the wrong text bellow Figure 1 "Figure 2. 6.2. SIM cards."
Thank you for your comment. We have removed the wrong text from the paper
d4) From page 9 until 13 it is very the same aspects comment in d2.
Thanks for your suggestion. The organization of the paper has been partially modified in order to facilitate its reading (see paper).
d5) Related work: It is important to mention other efforts, as it is stated in conclusion section as an affirmative text. This could be conceived before/after presenting the present contribution
Thanks for your suggestion. We have taken into account your suggestion in the new version of the paper
d6) Conclusions: two mentions could be considered:
- Re-write this section to better understanding the contribution, example is the first sentence:
"SMART4MD’s combination of a number of functions distinguishes it from other comparable 476 products and services, making it the kind of innovative non-pharmacological intervention that 477 experts in the field have called for [17]."
Probably consider: In this research work we proposed an approach, called as SMART4MD, which....
We appreciate your comment. We have modified the phrase as follows “In this research work we proposed an approach, called as SMART4MD, which combined of a number of functions distinguishes it from other comparable products and services, making it the kind of innovative non-pharmacological intervention that experts in the field have called for [17]”.
- Consider to insert Future Works, which could be done from you or other research groups.
We appreciate your comment. We have included the following sentence in the paper “Therefore, future studies could expand the number of participating countries, or propose a use of the app at home for a longer time”.
d7) e-Health, software engineering and hardware: this is a general comment to contribute to the authors to think about. This work has a nice contributions in its basis, or high level target to monitoring and provide a tool for Mild Dementia. Software engineering and hardware are not "the most important aspects". These could change, but the proposed idea no.
We appreciate your general comment and we totally agree with you, the most important thing about our SMART4MD app is not the software or the hardware, but the functionalities it provides for people with cognitive impairment / dementia, and for their caregivers. The purpose of using the app is to improve the quality of life of the people who use it.
Reviewer 2 Report
Point 1: The article presents useful information about feasibility study with MCU users, it is a study with potential real-world applications. Feasibility studies is a new challenge to break the gap between research and real life. I think qualitative information about the problems found implementing a technology is very interesting and an important information. In this study, two different countries have participated, and it is a valuable information. Despite that, a major revision is needed.
Point 2: I would expect that any study focused on usability and feasibility for individuals with MCI would provide more context about feasibility and usability studies (with or without MCI).Introduction must be improved.
The introduction is poor. It only dedicates the first 5 lines to the state of the art. The introduction should describe the problem to be solved, which studies have been carried out in the same direction (feasibility-usability), what the results were.
Given that feasibility hasn't the same method that usability study, I recommend that the authors provide information from these types of studies separately.
Some relevant references you can use to improve your introduction or justify your design:
Feasibility: Hermes, E. DA, Lyon, A. R., Schueller, S. M., & Glass, J. E. (2019). Measuring the Implementation of Behavioral Intervention Technologies: Recharacterization of Established Outcomes. Journal of Medical Internet Research, 21(1), e11752. https://doi.org/10.2196/11752
Iterative usability design (improving software during study): Castilla, D., Suso-ribera, C., Zaragoza, I., Garcia-palacios, A., & Botella, C. (2020). Designing ICTs for Users with Mild Cognitive Impairment : A Usability Study. International Journal of Environmental Research and Public Health, 17(5153), 1–21. https://doi.org/10.3390/ijerph17145153
Usability methods references: Maramba, I., Chatterjee, A., & Newman, C. (2019). Methods of usability testing in the development of eHealth applications: A scoping review. International Journal of Medical Informatics, 126, 95–104. https://doi.org/10.1016/j.ijmedinf.2019.03.018
Accessibility with MCI users: Haesner, M., Steinert, A., Lorraine O’sullivan, J., Steinhagen-Thiessen, E., & O’sullivan, J. L. (2015). Evaluating an accessible web interface for older adults-the impact of mild cognitive impairment (MCI). Journal of Assistive Technologies, 219–232. https://doi.org/10.1108/JAT-11-2014-0032
Point 3: Overall, an important problem of this draft is the lack of formal structure. The variables and measuring instruments and procedure is spread among all article sections.
Please, follow a formal structure for the article:
1. INTRODUCTION
2. METHOD
2.1. DESIGN: Describe at least briefly the design of your studio, not just describe it as "inspired by XXX study". You can provide information about usability design and feasibility design separately. Please, cite a relevant study to support your design. The study describen is a case study published in a journal non indexed in JCR database.
For feasibility design, see the well stablished study: Hermes, E. DA, Lyon, A. R., Schueller, S. M., & Glass, J. E. (2019). Measuring the Implementation of Behavioral Intervention Technologies: Recharacterization of Established Outcomes. Journal of Medical Internet Research, 21(1), e11752. https://doi.org/10.2196/11752
For usability studies you can see
2.2.PARTICIPANTS: Describe your inclusion criteria and sample. Number of subjects, socio-demographic characteristics, etc. Please, describe means, standard desviations for all the users (informal caregivers also).
2.3. MATERIALS:
2.3.1. VARIABLES AND MEASURING INSTRUMENTS (Cite the validation study of GDS and MMSE questionnaires, describe which constructs is measuring each questionnaire)
You could provide a table with column dyads (column 1 MCI Users, column 2 Informal caregivers users, column 3: place and conditions- home, hospital, etc). Line 1: session 1 measures (for each user), line 2: session 2 measures, etc., (including quantitative and qualitative variables and weekly calls). This table would help to clarify your design study.
2.3.2. HARDWARE
Describe all material used (included audio recording device)
2.3.3. SOFTWARE
Please describe enough the software used (each version).
2.3.4. Task description
Please describe here the user tasks. if the number of tasks is high, please summarize it in a table.
2.4. PROCEDURE
It would be desirable include a table with sessions schedule and a brief description of tasks for each session or group of sessions.
Do you give any advise to users and informal caregiver of how to use the system (training session, the order of tasks for this study, etc) or how do you control all the users do the same tasks?
Clarify if the MCI users should use
(sections 2.1. Recruitment, 2.2. Setup and timeline, 2.3. Ethical considerations and Informed consent, 2.4. Screening must be emplazed in procedure section)
3. RESULTS
3.1. Usability results
3.1.1. Quantitative results
Describe the results regarding measuring instruments described in 2.3.1. section.
3.1.2. Qualitative results
Here you can describe the opinion of users about tablets, software etc.
3.2. Feasibility results
3.3. Lessons learned
Describe here your findings and recommendations to improve the usability and feasibility.
4. DISCUSSION
Please, you should begin the section by taking up the objectives of this study, comparing its results with the previous literature described in the introduction. You have done an interesting feasibility study, the discussion should share your lessons learned alowing other researchers avoid the feasibility problems you found.
5. CONCLUSIONS
Highlight the main contributions of this work to the literature. Try to list some recommendationssummarizing your findings. Present real limitations.
Point 4: Some formal aspects of writing a scientific article are also missing (e.g. citing a table or figure in the text, explaining its content before putting it up). Type "table 4 presents the data ..."
Point 5: Line specific questions and corrections.
- Lines 49-66. This is not information from the introduction, but from the method. Please, re-write and emplace it in method section.
- Line 74. What id BTH?
- Line 79. What is CST?
- Line 84. "The main reason given was that the caregiver saw difficulties in 84 the use of the application by people with mild cognitive impairment (PwMCI)". How did you deal with this problem? What measures did you take to improve the acceptability of the application?
- Lines 108-114. Description of measures must be in variables and measuring instruments. Inclusion and exclusion criteria in participants section. In procedure section, only describes when, where, who and how apply this questionnaires.
- Line 110. MMSE is an screening tool, it is not enough to diagnose cognitive impairment. Please, explain in participants section if these users had a diagnosis of cognitive impairment before this study, by whom it was made (neurologist, geriatrician, etc.) and if they followed any protocols established by the health systems.
- Line 121. "first user testing session in June 2017" Please, describe the testing session, what tasks are involved in the session, what is the role of the MCI user, the rol of the informal caregiver, if there was an experimenter present during testing (or was he/she observing remotely), what was the role of the experimenter, etc.
- Line 123-124. "About 2 weeks after using the 122 initial version, the participants received an updated version of the app, which they were able to use for the remaining two weeks of the test". Please, describe in software section the two APP versions and why was needed to change software in the middle of the study."
- Lines 145-148. "Furthermore, a number of changes in the settings were made in order to improve accessibility and usability, such as horizontal/vertical lock, turning off screen lock, etc. A light blue background image was downloaded and set as the screen background". Plase, describe the original APP features, and describe adequately second version with improved features.
- Lines 150-153. Change to procedure section. Who did the installation? Usually, Android devices has desactivated the option to install APKs from out GooglePlay store. How did you avoid usability problems installing the APK?
- Lines 162-169. I can read between the lines that there was a training session with the tablet. Please, describe briefly the training session in the procedure section and the tasks performed in the 2.3.4. section (task description).
- Line 172. "and the Organic Law 15/1999 on 172 Personal Data Protection dated December 13 (Spain)" I am concerned that the European data protection regulation will not be cited for Spanish research, and I am also concerned that the Spanish law cited by the authors is not in vigour (In 2018 new law substitutes Law 15/1999 in order to meet the european 95/46/EC directive criteria).
- Line 173. "The audio recordings from the usability testing 173 sessions were recorded using voice recorders not connected to the internet". Please, include the audio devices in material (hardware section).
- Line 175. How the audio were transferred from devices to secure folders? By Internet? By hard drive?
- Line 183. "The only user data assessed in the feasibility study was the amount of SIM data/user/month". Please, describe this in measures section. Justify how this data can provide useful feasibility information.
- Lines 185-190. "Method for usability testing". The reference is not a well stablished study. Please provide well stablished references and describe enough the tasks and usability sessions.
- Lines 196-199. "According to this user testing method, users should not be given a guided introduction to the app before the testing session, since this might bias their opinions and result in less valuable feedback. Instead, they should be introduced to the app individually through the tasks they are asked to perform." Please, describe in variables and measuring instruments section if quantitative results from usability test were collected (number of attempts, successful or not task, etc). Maybe you cpould extrapolate some quantitative information from "Sections where the user struggled".
If you only collected qualitative opinion from users, maybe you should change your article title and adjust to work done "feasibility and user experience study". Maybe you could considere re-writing "Adapting the method to the conditions of the feasibility-usability study" section to fit with this suggestion too (lines 205-215). - LINEs 213-215. "than the recommended 6-10 were considered necessary. The aim was to receive feedback from at least 214 five users per task and site" Describe this information in the design section, not here.
- Line 224. "Ability to complete the task independently" How did you measure? Please describe this metric in variables and measuring instruments section.
- Line 254. Move the sample description to participants section.
- Lines 270-271. Describe the questions before procedure section in measuring instruments section.
- Line 300. This kind of tree must be move to procedure. It is not a result, it is a procedure explanation.
- Line 303-309. Move this description to participants sections description. Characteristics of your users is not a result of your study.
- Line 310 (Tablets) How did you avoid the usability issues related to the logical keyboard from tablets? It is well-stablished this is a problem for elderly users (much more for those with MCI).
- Line 320-321. Please describe briefly the problems found and why these features ("including horizontal/vertical lock, turning off screen lock, etc. A light blue background image was downloaded and set as the screen background") were included.
- Lines 322-323. "Email accounts had to be set up manually on each tablet as the the SMART4MD app needed to be installed through an APK file sent to these addresses" Please, clarify if it is a action taken due the usability results or if it was the initial configuration (if it was the inicial configuration please move to software description).
- Lines 330-339. "Phase I of the feasibility-usability study consisted of a short introduction followed by task-based 330 individual usability testing with first the PwMCI and then the carer (...)" This kind of desscription is not a result, is procedure. Please review all the article and unify all procedure descriptions in the procedure section.
- Lines 340-"User performance of tasks included in the usability testing"
Please try to inform some quantitative results. i.e. "A few PwMCI who were not familiar with tablets or smartphones were very insecure when approaching the tablet and app." How many users are "a few". What percentage of the sample represents? - Lines 406-417. Measures must be presented at variables and measuring instruments section. Here, in results you should present some kind of results. You can see how are explained the results in the studies I recommended at the beginning of this review.
- Line 430. What measures does "usability test scores" represent?
Author Response
Dear Reviewer
First of all, thank you for reviewing our paper. Next, we respond to each of your comments and suggestions:
Point 1: The article presents useful information about feasibility study with MCU users, it is a study with potential real-world applications. Feasibility studies is a new challenge to break the gap between research and real life. I think qualitative information about the problems found implementing a technology is very interesting and an important information. In this study, two different countries have participated, and it is a valuable information. Despite that, a major revision is needed.
Thank you very much for your comment. We will take into account all the recommendations-modifications that you propose in your review of our paper.
Point 2: I would expect that any study focused on usability and feasibility for individuals with MCI would provide more context about feasibility and usability studies (with or without MCI).Introduction must be improved.
The introduction is poor. It only dedicates the first 5 lines to the state of the art. The introduction should describe the problem to be solved, which studies have been carried out in the same direction (feasibility-usability), what the results were.
Given that feasibility hasn't the same method that usability study, I recommend that the authors provide information from these types of studies separately.
Some relevant references you can use to improve your introduction or justify your design:
Feasibility: Hermes, E. DA, Lyon, A. R., Schueller, S. M., & Glass, J. E. (2019). Measuring the Implementation of Behavioral Intervention Technologies: Recharacterization of Established Outcomes. Journal of Medical Internet Research, 21(1), e11752. https://doi.org/10.2196/11752
Iterative usability design (improving software during study): Castilla, D., Suso-ribera, C., Zaragoza, I., Garcia-palacios, A., & Botella, C. (2020). Designing ICTs for Users with Mild Cognitive Impairment : A Usability Study. International Journal of Environmental Research and Public Health, 17(5153), 1–21. https://doi.org/10.3390/ijerph17145153
Usability methods references: Maramba, I., Chatterjee, A., & Newman, C. (2019). Methods of usability testing in the development of eHealth applications: A scoping review. International Journal of Medical Informatics, 126, 95–104. https://doi.org/10.1016/j.ijmedinf.2019.03.018
Accessibility with MCI users: Haesner, M., Steinert, A., Lorraine O’sullivan, J., Steinhagen-Thiessen, E., & O’sullivan, J. L. (2015). Evaluating an accessible web interface for older adults-the impact of mild cognitive impairment (MCI). Journal of Assistive Technologies,219–232. https://doi.org/10.1108/JAT-11-2014-0032
Thank you for your comment. We have expanded the introduction as you suggested (see paper)
Point 3: Overall, an important problem of this draft is the lack of formal structure. The variables and measuring instruments and procedure is spread among all article sections.
Please, follow a formal structure for the article:
- INTRODUCTION
- METHOD
2.1. DESIGN: Describe at least briefly the design of your studio, not just describe it as "inspired by XXX study". You can provide information about usability design and feasibility design separately. Please, cite a relevant study to support your design. The study describen is a case study published in a journal non indexed in JCR database.
For feasibility design, see the well stablished study: Hermes, E. DA, Lyon, A. R., Schueller, S. M., & Glass, J. E. (2019). Measuring the Implementation of Behavioral Intervention Technologies: Recharacterization of Established Outcomes. Journal of Medical Internet Research, 21(1), e11752. https://doi.org/10.2196/11752
For usability studies you can see
2.2.PARTICIPANTS: Describe your inclusion criteria and sample. Number of subjects, socio-demographic characteristics, etc. Please, describe means, standard desviations for all the users (informal caregivers also).
2.3. MATERIALS:
2.3.1. VARIABLES AND MEASURING INSTRUMENTS (Cite the validation study of GDS and MMSE questionnaires, describe which constructs is measuring each questionnaire)
You could provide a table with column dyads (column 1 MCI Users, column 2 Informal caregivers users, column 3: place and conditions- home, hospital, etc). Line 1: session 1 measures (for each user), line 2: session 2 measures, etc., (including quantitative and qualitative variables and weekly calls). This table would help to clarify your design study.
Incluir en measures
2.3.2. HARDWARE
Describe all material used (included audio recording device)
2.3.3. SOFTWARE
Please describe enough the software used (each version).
2.3.4. Task description
Please describe here the user tasks. if the number of tasks is high, please summarize it in a table.
2.4. PROCEDURE
It would be desirable include a table with sessions schedule and a brief description of tasks for each session or group of sessions.
Do you give any advise to users and informal caregiver of how to use the system (training session, the order of tasks for this study, etc) or how do you control all the users do the same tasks?
Clarify if the MCI users should use
(sections 2.1. Recruitment, 2.2. Setup and timeline, 2.3. Ethical considerations and Informed consent, 2.4. Screening must be emplazed in procedure section)
- RESULTS
3.1. Usability results
3.1.1. Quantitative results
Describe the results regarding measuring instruments described in 2.3.1. section.
3.1.2. Qualitative results
Here you can describe the opinion of users about tablets, software etc.
3.2. Feasibility results
3.3. Lessons learned
Describe here your findings and recommendations to improve the usability and feasibility.
- DISCUSSION
Please, you should begin the section by taking up the objectives of this study, comparing its results with the previous literature described in the introduction. You have done an interesting feasibility study, the discussion should share your lessons learned alowing other researchers avoid the feasibility problems you found.
- CONCLUSIONS
Highlight the main contributions of this work to the literature. Try to list some recommendationssummarizing your findings. Present real limitations.
Thanks for your recomendation. We have followed the proposed structure (see paper).
Point 4: Some formal aspects of writing a scientific article are also missing (e.g. citing a table or figure in the text, explaining its content before putting it up). Type "table 4 presents the data ..."
Thanks for your suggestion. We have included a previous text before including the figures or tables (see paper).
Point 5: Line specific questions and corrections.
- Lines 49-66. This is not information from the introduction, but from the method. Please, re-write and emplace it in method section.
Thanks for your suggestion. We have included this text in the method
- Line 74. What id BTH?
The initials BTH stand for Blekinge Institute of Technology (Swedish initials - BTH). In the introduction section it was indicated. Although this section has moved to the method section, the first time BTH appears in the article the meaning of the acronyms is explained
- Line 79. What is CST?
The initials CST stand for Consorci Sanitari de Terrassa. In the introduction section it was indicated. Although this section has moved to the method section, the first time CST appears in the article the meaning of the acronyms is explained
- Line 84. "The main reason given was that the caregiver saw difficulties in 84 the use of the application by people with mild cognitive impairment (PwMCI)". How did you deal with this problem? What measures did you take to improve the acceptability of the application?
Thanks for your comment. The assessment of whether the subject was "fit" to use the application was made by the caregiver himself, therefore, it was a subjective assessment. We think that more than the use of the application itself, they were based on the use of the tablet or a technological tool. In the case of Spain, many older subjects do not have much experience in the use of mobile applications or tablets. Therefore, in the recruitment phase, we do not propose to face this problem of "acceptability perceived by the caregiver"
- Lines 108-114. Description of measures must be in variables and measuring instruments. Inclusion and exclusion criteria in participants section. In procedure section, only describes when, where, who and how apply this questionnaires.
Thank you for your comment. The description of the instruments is included in the corresponding section
- Line 110. MMSE is an screening tool, it is not enough to diagnose cognitive impairment. Please, explain in participants section if these users had a diagnosis of cognitive impairment before this study, by whom it was made (neurologist, geriatrician, etc.) and if they followed any protocols established by the health systems.
Thank you for your comment. Strongly agree, the MMSE is a screening test not a diagnostic test. The new version of the paper describes the inclusion and exclusion criteria in more detail. For this usability-feasibility study, the same inclusion criteria were used as for the clinical trial. In reference to the cognitive state of the subject:
- Participants score 20 to 28 points on the MMSE whether or not a diagnosed neurodegenerative disease is present
- A professional assessment of the patient's own experience of memory problems over a substantial period of time (more than 6 months)
- Line 121. "first user testing session in June 2017" Please, describe the testing session, what tasks are involved in the session, what is the role of the MCI user, the rol of the informal caregiver, if there was an experimenter present during testing (or was he/she observing remotely), what was the role of the experimenter, etc.
Thanks for you comment. As described later in the article (see section Feasibility-usability study phase I), in this testing session “When a dyad arrived at the introduction session, they were first given general information about the setup of the feasibility-usability study as a whole followed by more specific information about the introductory meeting. They were also informed and asked to give their consent to having the session audio recorded. Usability testing was then done individually with first the PwMCI and then the carer. While one person in the dyad did the usability testing, the other person was asked to wait outside. Since not all participants had time to complete all the tasks within the timeframe, the tasks were divided up so that each task would be tested by at least five participants. The most central tasks, which were part of the ‘reminder section’ of the app and which had to be explained to all users, were tested by more participants than the other tasks. When both persons in the dyad had completed the usability testing, the test leader met with them together and answered any questions that they had about the tablet, app or the study”.
Both the subject with MCI and the caregiver had the opportunity to test the application during the session, this test was done as it has been described in an individual way. The experimenter had the role of directing the training / testing session of the application. First, I would do an explanation of the application. Then he showed the subject or the caregiver how the application was used, and then asked both of them to carry out some tasks.
- Line 123-124. "About 2 weeks after using the 122 initial version, the participants received an updated version of the app, which they were able to use for the remaining two weeks of the test". Please, describe in software section the two APP versions and why was needed to change software in the middle of the study."
Thanks for your comment. The project management and the clinical lead in this project decided that it was necessary to start the introduction and user testing of the app by latest early June 2017 in order not to delay the project further. Due to the delay of the app development it was decided that within the frames of the existing project, try to compromise the feasibility study in order not to delay the SMART4MD project further. A two-week period had originally been scheduled in between the initial usability testing sesión and the handing out the app. During this time, feedback from the first sesión was supposed to be collected to App developer and implemented into a new updated version, that was to be handed out to the participans. Considering that App developer were still working on implimenting the original functionality, this intermediate time was deleted. Instead participants who participated in the first user testing sesión were provided with the early version of the app directly so that their four weeks of usage could start immediately and not have to wait until the new version of the app had been developed. About two weeks into using the first version, participants were then invited to receive un updated version of the app, which they were tan able to test for the remaining two weeks of the test period.
- Lines 145-148. "Furthermore, a number of changes in the settings were made in order to improve accessibility and usability, such as horizontal/vertical lock, turning off screen lock, etc. A light blue background image was downloaded and set as the screen background". Plase, describe the original APP features, and describe adequately second version with improved features.
Thank you for your comment. This section describes the adjustments made in the tablets to improve their accessibility and usability, they do not refer to the APP features.
Specifically, the tablets were prepared as follows:
Preparing the tablet for installation of app
- Charge tablet
- Fixed portrait orientation
Settings > Display > When device is rotated > Stay in current orientation
- Sleep after 30 minutes of inactivity
Settings > Display > Sleep > 30 minutes
- Remove screen lock
Settings > Security > Screen lock > None
- Desktop is blank
(SMART4MD launcher only on screen, all other apps/widgets removed from desktop)
- Remove non-essential apps
- Disable notifications from all other apps (including games)
- Lines 150-153. Change to procedure section. Who did the installation? Usually, Android devices has desactivated the option to install APKs from out GooglePlay store. How did you avoid usability problems installing the APK?
Thanks for your suggestion. This section change to procedure section. The SMART4MD app was sent as an apk file, first from developer to the technicians at CST and BTH, who were responsible for installing the app on each of the tablets. Then from the technicians to the different e-mail address of the individual tablets, afrom were the app was installed.
- Lines 162-169. I can read between the lines that there was a training session with the tablet. Please, describe briefly the training session in the procedure section and the tasks performed in the 2.3.4. section (task description).
Thanks for your suggestion. We have included such information in the procedure section
- Line 172. "and the Organic Law 15/1999 on 172 Personal Data Protection dated December 13 (Spain)" I am concerned that the European data protection regulation will not be cited for Spanish research, and I am also concerned that the Spanish law cited by the authors is not in vigour (In 2018 new law substitutes Law 15/1999 in order to meet the european 95/46/EC directive criteria).
Thank you for your comment. This study was presented and approved before 2018, hence the reason for including the previous law. The amendments to the project include the new law. In Spain we use European law
- Line 173. "The audio recordings from the usability testing 173 sessions were recorded using voice recorders not connected to the internet". Please, include the audio devices in material (hardware section).
Thanks for your suggestion. We have included this information in the hardware section. It is important to note that these audio recordings were not made with the project's tablets, if not, with other devices.
- Line 175. How the audio were transferred from devices to secure folders? By Internet? By hard drive?
The audios were used to transfer the comments of the participants in the usability session. They were not transferred to the devices.
- Line 183. "The only user data assessed in the feasibility study was the amount of SIM data/user/month". Please, describe this in measures section. Justify how this data can provide useful feasibility information.
This data from the use of SIM data was relevant for deciding what would be the limit to be established in the clinical trial. In this usability-feasibility study the average usage was 641 MB / user / month. However, in the feasibility study several updates of the app were made and games were downloaded using the SIM. In the clinical trial, where the amount of SIM data limited to 60 MB / user / month, WiFi will be used for these updates and downloads.
- Lines 185-190. "Method for usability testing". The reference is not a well stablished study. Please provide well stablished references and describe enough the tasks and usability sessions.
Thank you for your comment. The usability method has been described in detail in the corresponding section
- Lines 196-199. "According to this user testing method, users should not be given a guided introduction to the app before the testing session, since this might bias their opinions and result in less valuable feedback. Instead, they should be introduced to the app individually through the tasks they are asked to perform." Please, describe in variables and measuring instruments section if quantitative results from usability test were collected (number of attempts, successful or not task, etc). Maybe you cpould extrapolate some quantitative information from "Sections where the user struggled".
If you only collected qualitative opinion from users, maybe you should change your article title and adjust to work done "feasibility and user experience study". Maybe you could considere re-writing "Adapting the method to the conditions of the feasibility-usability study" section to fit with this suggestion too (lines 205-215).
Thank you for your comment. As described in the article, both quantitative and qualitative variables were collected.
In the usability testing the following quantitative metrics on efficacy and efficiency were collected:
- Ability to complete the task independently
- Time to complete the tasks (seconds)
- Need for guidance (by caregiver or clinical test leader) to complete the task
In addition, subjective metrics were collected, including questions after each completed task. The qualitative metrics were the following:
- Sections where the user struggled
- Difficulties experienced by the user when completing the task
- Features missing that would substantially help according to the dyad
These measurements will be described in the variable and measuring instruments section.
- LINEs 213-215. "than the recommended 6-10 were considered necessary. The aim was to receive feedback from at least 214 five users per task and site" Describe this information in the design section, not here.
Thanks for your suggestion. It has been taken into account in the new organization of the paper.
- Line 224. "Ability to complete the task independently" How did you measure? Please describe this metric in variables and measuring instruments section.
Thank you for your comment. The ability to complete the task independently was measured when the task was completed autonomously, that is, the user did not require the help of the evaluator. This measurement has been described in the measuring instruments section.
- Line 254. Move the sample description to participants section.
Thanks for your suggestion. We have followed your recommendation (see paper)
- Lines 270-271. Describe the questions before procedure section in measuring instruments section.
Thanks for your suggestion. We have followed your recommendation (see paper)
- Line 300. This kind of tree must be move to procedure. It is not a result, it is a procedure explanation.
Thanks for your suggestion. The figure has been included in the procedure
- Line 303-309. Move this description to participants sections description. Characteristics of your users is not a result of your study.
Thanks for your suggestion. Characteristics of the users has been included in the participants section.
- Line 310 (Tablets) How did you avoid the usability issues related to the logical keyboard from tablets? It is well-stablished this is a problem for elderly users (much more for those with MCI).
Thank you for your comment. This problem was not solved in this study. But it will be taken into account for future studies.
- Line 320-321. Please describe briefly the problems found and why these features ("including horizontal/vertical lock, turning off screen lock, etc. A light blue background image was downloaded and set as the screen background") were included.
Thank you for your comment. These features were defined prior to conducting the study. These settings were selected to make the use of the tablet easier, as well as to homogenize the settings across all tablets.
- Lines 322-323. "Email accounts had to be set up manually on each tablet as the the SMART4MD app needed to be installed through an APK file sent to these addresses" Please, clarify if it is a action taken due the usability results or if it was the initial configuration (if it was the inicial configuration please move to software description).
Thank you for your comment. This is an initial setup. This section explains the difficulties of doing it in this way and the decisions, for example, to include the MDM platform in the full pilot
- Lines 330-339. "Phase I of the feasibility-usability study consisted of a short introduction followed by task-based 330 individual usability testing with first the PwMCI and then the carer (...)" This kind of desscription is not a result, is procedure. Please review all the article and unify all procedure descriptions in the procedure section.
Thanks for your suggestion. This suggestion has been taken into account in the new organization of the paper
- Lines 340-"User performance of tasks included in the usability testing"
Please try to inform some quantitative results. i.e. "A few PwMCI who were not familiar with tablets or smartphones were very insecure when approaching the tablet and app." How many users are "a few". What percentage of the sample represents?
Thank you for your comment. The subjects were asked what was the frequency of use of Smartphone or tablet. To this question, 9 of the 10 PwMCI answered that "they had never used a smartphone or tablet". Three of these 9 subjects were from the Swedish sample (BTH) while the remaining six were from the Spanish sample (CST).
- Lines 406-417. Measures must be presented at variables and measuring instruments section. Here, in results you should present some kind of results. You can see how are explained the results in the studies I recommended at the beginning of this review.
Thank you for your comment. Your suggestion has been taken into account in the new organization of the paper
- Line 430. What measures does "usability test scores" represent?
Thank you for your comment. This table presents the data of the user evaluation scores, specifically, the summation of the assessment of each of the assessed attributes of the SMART4MD app is presented. In addition, in this table we have included the assessment of the different attributes in the user evaluation. It has been clarified in the paper
Round 2
Reviewer 1 Report
Dear Authors,
In this new version of the research work it is possible to see a large effort in enhancing aspects suggested related to the previous document.
I would like "only" to reinforce, if you do not accept it is alright to me, is that in my point of view referring to the title, which could be "only":
"Support Monitoring and Reminder Technology for Mild Dementia "
This comment is based upon two reasons: (1) the importance of your focus in the mentioned suggested title and (2) it will prove a more "long life" of this research in terms of systematic search from other future searches.
Therefore, I agree that this work is in the level to be publish.
Author Response
Thank you very much for your second review, as well as the suggested title change. The reason for specifying the type of study within the SMART4MD project is that the study protocol has been published first. We have also published results of some of the tests administered. We plan to publish the final results of the clinical trial, therefore, it seems relevant to us to specify that it is the feasibility-usability study within the SMART4MD project. Again, we appreciate your suggestion.
Reviewer 2 Report
Point 1: General comments:
1.1. The new procedure section and measures have improved the article significantly, but it has still some problems in the formal structure, including some aspects of the procedure in sections that do not correspond.
1.2.In the results section are missing some relevant data. Given that you define in your title your study as a "feasibility study" and in measures only gather "the amount of SIM data/user/month" regarding feasibility, the results section should show specific data about it, and explain in discussion why is important (you facilitate this information in your cover letter with answers to the revision). Also, cultural perspective is introduced as an important aspect by which the study is carried out in different countries, but you not show any result linked to the diferences or similarities between both cultures. Please, provide some specifical data about this question.
1.3. Finally, the results are inaccurately described.
On the one hand, the general results are described as imprecise data "Several PwMCI said...", "A few PwMCI who were not familiar with tablets or smartphones were very insecure", "Others approached the technology with curiosity", "The cognitive games included in the app were appreciated", "A paper-based manual was seen by several dyads". Please, provide precise data, for example "20% of PwMCI reported..." Please, revise all your results sections and provide accurate results such as frequencies of succesful task, mean and SD of number of task done by the users, etc.
On the other hand, an important score (summation score) is presented, but it is not justified with theoretical references.
-----------------
Point 2: Line specific questions and corrections.
Lines 68-70. Organization as “the effectiveness, efficiency, and satisfaction with which specified users can achieve goals in particular environments”
Please provide cite.
The introduction have improve but is still brief. The examples:
A introduction should answer questions like: What is the problem? Why is important? What constructs are involved in this problem? You are talking about cultural differences (that's your design leitmotiv and the reason to carry out your trial in several countries), the importance of a user centered design, the feasibility of this kind of projects...etc. What studies have been conducted in the literature?(the 4 that I provided were examples, but not enough to contextualize the study).
What did these studies contribute and what things were left unanswered?
Your study is covering a gap in the literature. Introduction must facilitate context to understand the concepts used in the paper, and highlight the gap that this study will cover. After in discussion you should try to bring back the introduction and explain how your work cover these gaps.
Please, provide a more strong background (including references from reliable scientific database like Journal Citation Report, scopus, PubMed, etc).
Line 169 "Participants score 20 to 28 points on the Mini Mental State Examination (MMSE) whether".
Although the cut-off point in the MMSE could be anywhere between 1-30, the score of 24 is normally used to discriminate possible impairment. In the case of a higher score, how have the authors ensured that the participants have mild dementia? I understand (due the information of procedure section) that potencial patients were located from primary care database, so I understand there is a previous diagnoses. If a previous diagnoses is needed, the authors should describe in inclusion criteria and explain briefly in procedure.
Line 285 (empty). If a diagnoses is needed, pleas, add a sentence after your 28 point score explanation, emphasizing that you are sure of its validity due to the previous diagnosis.
Lines 190-211. That information is not participants. It must be on procedure section.
Line 199 "Site BTH is localted in Terrassa (Barcelona). At CST, potential" Maybe is an error and authors means "Site CST".
Line 201. In a peer-review journal is not acceptable to show any name of the authors in the paper, in order to avoid any possible bias in review process. Please anonymize using professional status intead of the name (the medical staff in charge, the professional staff, etc).
Lines 213-226. The description of the sociodemographic data of the sample is still missing in this section. Mean age and standard deviation, percentage of men and women with cognitive impairment, etc.(described on lines 256... must be placed here).
Timeline is a procedure information, not a particpants section information.Please, move to procedure section.
Line 253. Flowchart must be placed in procedure section.
Line 270. You have improve remarkably this section. Thank you for clarify.
Line 310. You can strong your validity data referencing Hermes article.
Lines 493-499. Please provide some context and evidence of this kind of tools in the introduction section.
Line 524. Given that is not acceptable refer name of the researchers in a scientific paper, please change the sentence anonymizing the names.
Line 540-541. Given that is not acceptable refer name of the researchers in a scientific paper, please change the sentence anonymizing the names.
Line 646-650. It is procedure information, not results. Please, move the paragraph to procedure section.
Line 651. "Since not all participants had time to complete all the tasks" Given that succesful task and time are important variables of usability measures, please, provide data with percentage of users who did not complete the all the task and time.
Lines 651-654- "timeframe, the tasks were divided up so that each task would be tested by at least five participants. The most central tasks,which were part of the ‘reminder section’ of the app and which had to be explained to all users, were tested by more participants than the other tasks" That's no results information, but it 's procedure. Please, move to correct section.
Lines 682-691. "....measure was the SUS". This is not results but procedure. Please move to procedure section.
Lines 691-697. "However, the SUS did.... questions asked". This is not results. The hypotheses that explain the reason for the results must be placed in discussion. That's a formal structure for scientific articles.
Lines 705-707 "However, as some new functionalities and updates had not yet been fully tested, a decision was made to have a complementary round of usability testing and evaluation as soon as the remaining parts of the app were fully functional". This is not results but procedure. Please move to procedure section.
Lines 726-730 "Before phase I of the feasibility-usability study, the tablets were prepared in such a way that any external apps were deleted from the home screen in order to make interaction with the SMART4MD app simpler for the target group. Furthermore, a number of settings were changed in order to improve accessibility and usability, including horizontal/vertical lock, turning off screen lock, etc. A light blue background image was downloaded and set as the screen background."
It is not clear yet what is new and what is procedure. Please revise this paragraph and clarify wich changes are needed as a result of the study.
Lines 746-748. It's the first time in the manuscript that a table pens are mentioned. Please, describe it at materials section/hardware.
Lines 759 onwards. Discussion section.
Discussion section try to bring some reflections and explanations to the results. A formal discussion must compare its results with literature cited in the introduction, and to close the question about what this study has contributed to the gap in the literature, limitations and new questions for the future work.
Line 840-841 "Recruitment was successful as evidenced by the acceptance rate of the participants when the"
That's righ, but you must introduce this result on results section before explain it in the discussion section. Please, try to explain the important difference between Sweden and Spain cases in the discussion.
Lines 859-860- "Our results indicate that familiarity with similar technology varies strongly among participants."
Did you gather ICT experience profile of your participants? This conclusion is relevant but
Lines 832-833. "The purpose of conducting the feasibility study in two different countries was to examine whether any linguistic and cultural differences impacted"
You don't explain cultural differences despite it is a core in your design, but it is not supported by any concrete data from your study. Please provide a subsection in results section about differences between both countries. After explain the differences in discussion section.
Lines 864-866. "A few PWDs who were not familiar with tablets or smartphones were very unsecure when approaching 865 the tablet and app." Specify this data before in the results section.
Lines 868-870 "However, the majority think that the use of smartphones or tablets is helpful for memory and this highlights the presence of a gap between the perceived potential and the actual use of these technologies" That's a qualitative result, not discussion.
line 935 "Furthermore, the study carried out in two countries Sweden and Belgium". It's must be an error. Did you want to say Sweden and Spain?
---------------------------------------
POINT 3: Author responses Review 1
Answer to "Line 84.
"The main reason given was that the caregiver saw difficulties in 84 the use of the application by people with mild cognitive impairment (PwMCI)". How did you deal with this problem? What measures did you take to improve the acceptability of the application? Thanks for your comment. The assessment of whether the subject was "fit" to use the application was made by the caregiver himself, therefore, it was a subjective assessment. We think that more than the use of the application itself, they were based on the use of the tablet or a technological tool. In the case of Spain, many older subjects do not have much experience in the use of mobile applications or tablets. Therefore, in the recruitment phase, we do not propose to face this problem of "acceptability perceived by the caregiver"
It could be interesting you explain these differences in your discussion.
You answer to Line 121 suggestion is very clear (thank you). Please, check you have explained as well in the paper.
Answer to lines 145-148. This is a very important information to be able to replicate your study. Please provide in material or in additional appendix.
Answer to lines 150-153. Your answer is very clear in the cover letter but not in the paper. Please explain it in the same way in procedure, so that the reader knows that this installation was made by the technicians.
Answer to line 173. "Thanks for your suggestion. We have included this information in the hardware section. It is important to note that these audio recordings were not made with the project's tablets, if not, with other devices." This information is still confusing. Please add the adjetive "external" to help to understand this concept. "An external audio recorder was used for..."
Answer to Line 253. Regarding next cover letter explanation "About two weeks into using the first version, participants were then invited to receive un updated version of the app, which they were tan able to test for the remaining two weeks of the test period". I fully understand the needs of real time projects, but you must be very accurate describing your study, given that papers represent the "recipe" of science. So, please explain in a clear way who did what. Please, include in your flowchart how many dyads have used 1 version and how many have used 2 different versions.
How many users used both softwares? How many users used only sencond one? Maybe you can describe the study like a iterative process (and justify with any reference) to explain why you are changing the experimental conditions during the study.
Answer to line 310. Given that the study has been done with 7" tablets and elderly users, it seems unbelievable you didn't found any comment about this common problem. The most common solution is to facilitate a phisical keyboard with the tablet to avoid it.
Answer to Line 430. What measures does "usability test scores" represent? "Thank you for your comment. This table presents the data of the user evaluation scores, specifically, the summation of the assessment of each of the assessed attributes of the SMART4MD app is presented. In addition, in this table we have included the assessment of the different attributes in the user evaluation. It has been clarified in the paper".
It is not clear enough. If this score is a summation of all scores, you are considering all the measures like a big construct and as far I know, it has no theoretical support. If it has, please provide in introduction and retake in the discussion. In v2 manuscript, you describe in Lines 699-700 "User evaluation was analysed in the following way: if a respondent had a minimum total score of 60% (15 out of 25) or more, he or she was considered to be satisfied with the application".
If this summation has not theoretical support, you should explain your measures one by one, not as a sum, and in your discussion you can give to the reader an overview commenting these positive results.
